# Efficient Transendothelial Migration of Latently HIV-1-Infected Cells

**DOI:** 10.3390/v13081589

**Published:** 2021-08-11

**Authors:** Reou Tanabe, Yuko Morikawa

**Affiliations:** Graduate School of Infection Control Sciences, Kitasato University, Shirokane 5-9-1, Minato-ku, Tokyo 108-8641, Japan; mi18009@st.kitasato-u.ac.jp

**Keywords:** transmigration, lymphocyte homing, reconstitution, HIV-1, latent infection, stromal cells

## Abstract

A small fraction of HIV-1-infected T cells forms populations of latently infected cells when they are a naive T-cell subset or in transit to a resting memory state. Latently HIV-1-infected cells reside in lymphoid tissues and serve as viral reservoirs. However, whether they systemically recirculate in the body and re-enter the lymphoid nodes are unknown. Here, we employed two in-vitro cell coculture systems mimicking the lymphatic endothelium in lymph nodes and investigated the homing potential, specifically the transendothelial migration (TEM), of two latently HIV-1-infected cell lines (J1.1 and ACH-2). In trans-well coculture systems, J1.1 and ACH-2 showed higher TEM efficiencies than their parental uninfected and acutely infected cells. The efficiency of TEM was enhanced by the presence of stromal cells, such as HS-5 and fibroblastic reticular cells. In an in-vitro reconstituted, three-dimensional coculture system in which stromal cells are embedded in collagen matrices, J1.1 showed slightly higher TEM efficiency in the presence of HS-5. In accordance with these phenotypes, latently infected cells adhered to the endothelial cells more efficiently than uninfected cells. Together, our study showed that latently HIV-1-infected cells enhanced cell adhesion and TEM abilities, suggesting their potential for efficient homing to lymph nodes.

## 1. Introduction

Combination antiretroviral therapy (cART) targeting HIV-1 replication can reduce the plasma viral loads in HIV-1-infected patients to undetectable levels. However, HIV-1 cannot be eradicated because of latent HIV-1 reservoirs [1,2], which are established at the early stage of infection. The majority of cells infected with HIV-1 become virus-producing cells and are rapidly destroyed. However, naive cells and memory T cells are relatively resistant to HIV-1 replication and often form latently infected cell populations. These cells harbor an integrated viral genome (provirus) but do not express HIV-1 genes. When HIV-1-producing T cells transit to a resting phase, viral gene expression from the provirus is also silenced, forming a memory T-cell population that harbors latent HIV-1 [3,4]. Latency appears to be a means of extending the survival of HIV-1 through immune evasion in vivo because latently infected cells are ostensibly indistinguishable from uninfected T cells.

Lymphoid tissues, more specifically gut-associated lymphoid tissues and lymph nodes, are the primary targets for HIV-1 infection [5]. The gut is the earliest target of HIV-1 infection, and it has been estimated to harbor 1 × 10^9^ infected T cells [6]. Lymph nodes have been estimated to serve approximately 50% of the plasma viral load, which can be reduced to <1% by cART [7]. However, HIV-1 RNA/DNA can still be detected after years of cART [8], suggesting that latently infected cells harboring replicative or inducible HIV-1 reside in the lymphoid tissues and can recirculate in the blood, leading to systemic latent infection [9].

Naive lymphocytes circulate in the body and return to the lymph nodes from the blood. When lymphocytes reach specialized high endothelial venules (HEV) in the lymph nodes, they bind to the blood vessels and migrate into the lymph nodes. This process, termed lymphocyte homing, is composed of multiple, sequential steps that are regulated by the interactions between integrins expressed on lymphocytes and their ligands expressed on HEV [10,11,12]: (i) lectin-like leukocyte adhesion protein (L-selectin) expressed on lymphocytes weakly interacts with peripheral node addressins (PNAd) present on HEV, which leads to lymphocyte rolling and arrest on HEV [13]; (ii) chemokines stimulate lymphocytes by inside-out signaling and induce conformational changes in lymphocyte function-associated antigen-1 (LFA-1) [14,15]; (iii) activated LFA-1 tightly binds to intercellular adhesion molecule-1 (ICAM-1) on HEV; and (iv) finally, lymphocytes become polarized and extravasate from the blood vessels to the tissues. It is also known that LFA-2 (CD2) of T cells interacts with LFA-3 (CD58) on endothelial cells [16]. VLA-4 expressed on leukocytes interacts with VCAM-1 and recruits leukocytes to inflammatory sites. PECAM-1 is expressed on endothelial cells as well as leukocytes and mediates leukocyte transmigration [17]. Since latently HIV-1-infected cells are phenotypically similar to uninfected cells, it is possible that they may recirculate in the blood and re-enter the lymphoid tissues. This possibility has been suggested in mouse models injected with HIV-1-infected cells [18]. However, such studies in human subjects have not been reported, and homing, that is, transendothelial migration (TEM) of latently infected or even productively infected cells, has not been studied.

Lymph nodes are highly structured tissues in which several types of stromal cells form a network that partitions T and B lymphocytes [19,20]: follicular dendritic cells (FDC) and marginal reticular cells (MRC) in the B-cell-rich area (cortical area) produce CXC-chemokine ligand (CXCL) 12 and CXCL13, whereas fibroblastic reticular cells (FRC) in the T-cell-rich area (paracortical area) produce CC-chemokine ligand (CCL) 19 and CCL21. Naive T cells express CC-chemokine receptor 7 (CCR7) for its ligands CCL19 and CCL21 and CXC chemokine receptor 4 (CXCR4) for its ligand CXCL12. These ligand-receptor interactions play pivotal roles in lymphocyte differentiation, migration, and homeostasis. These chemokines are also delivered to the luminal surface of the HEV and support the TEM of lymphocytes [12]. It is well known that HIV-1 uses CXCR4 as a coreceptor for viral entry, and thus the entry can be antagonized by CXCL12. Interestingly, recent studies have suggested that treatment with CCL19 and CCL21 promotes the establishment of HIV-1 latency in primary resting T lymphocytes [21,22]. These findings suggest a possible link of chemokine-mediated lymphocyte homing and maintenance of the HIV-1 reservoir cell population in lymphoid tissues.

To understand the homing of latently HIV-1-infected cells, it is necessary to establish in-vitro cell systems that mimic the HEV in the lymph node microenvironment. In this study, using two latent HIV-1-infected T cell lines (J1.1 and ACH-2), the adhesive abilities to endothelial cells and the efficiencies of TEM were explored. We further developed an in-vitro reconstituted, three-dimensional (3-D) lymphatic endothelium system using a stromal cell-embedded collagen gel. Quantitative analysis of TEM efficiency in these cell systems could be of value for evaluating the homing/maintenance efficiency of the HIV-1 reservoir in lymph nodes.

## 2. Materials and Methods

### 2.1. Cell Culture

J1.1 and ACH-2, derivatives of Jurkat and A3.01 T cell lines in which the HIV-1 LAV strain was latently infected, respectively [23,24], were obtained from the NIH AIDS Reagent Program and were grown in RPMI1640 medium supplemented with 10% fetal bovine serum (FBS). The parental Jurkat and A3.01 cell lines were obtained from American Type Culture Collection (ATCC) and were similarly grown in RPMI1640 medium. EA.hy926, a human umbilical vein endothelial cell (HUVEC)-derived endothelial cell line [25], and HS-5, a bone-marrow-derived stromal cell line [26], were from ATCC and were grown in Dulbecco’s modified Eagle’s medium (DMEM) with 10% FBS. HS-27A, another bone-marrow-derived stromal cell line (ATCC), was grown in RPMI1640 with 10% FBS. Primary FRC (ScienCell Research Laboratories) were grown in fibroblast medium (ScienCell Research Laboratories) supplemented with 1% fibroblast growth supplement and 2% FBS.

### 2.2. Virus and Infection

HIV-1 (the NL43 strain) containing the vesicular stomatitis virus (VSV) G protein was produced by co-transfection of the HIV-1 molecular clone pNL34 and the VSV-G protein-expression plasmid pHCMV-VSV-G into Lenti-X 293T cells. A pNL43 derivative expressing Gag-iEGFP was constructed as described previously [27,28]. The EGFP gene was cloned between the matrix/p17 and capsid/p24 genes in pNL43. This pNL43 derivative was similarly co-transfected with pHCMV-VSV-G into Lenti-X 293T cells to generate EGFP-expressing HIV-1 containing VSV-G protein. Production of HIV-1 in the culture medium was measured using an HIV-1 p24 antigen capture ELISA kit (Zeptometrix, Baffalo, NY, USA). For acute infection, Jurkat and A3.01 cells were infected with HIV-1 at a multiplicity of infection (MOI) of 2 and cultured at 37 °C for 2 days.

### 2.3. Cell Adhesion Assay

Latently infected J1.1 and ACH-2 cells and acutely infected Jurkat and A3.01 cells (at 2 days post-infection) were labeled with 2.0 µM CellTracker CMFDA (Thermo Fisher Scientific, Waltham, MA, USA). EA.hy926 cells were seeded in fibronectin-coated 12-well plates (1 × 10^5^ cells/well) and incubated for 2 days before coculture. One day before coculture, EA.hy926 cells were stimulated with 10 ng/mL TNF-α (Sigma-Aldrich, St. Louis, MO, USA) overnight. After washing with fresh medium, the labeled latently or acutely infected cells (5 × 10^5^ cells per well) were added to the EA.hy926 cell monolayer and cocultured in prewarmed RPMI1640 containing 50 ng/mL CCL19 (BioLegend, San Diego, CA, USA), CCL21 (BioLegend), and CXCL12 (Sino Biological, Beijing, China) at 37 °C for 1 h. Uninfected Jurkat and A3.01 cells were used in parallel. The cells were washed with prewarmed phosphate-buffered saline (PBS) and lysed with radioimmunoprecipitation (RIPA) buffer (Nacalai Tesque, Kyoto, Japan). The mean fluorescence intensity (MFI) in the lysate was measured using a microplate reader at 480 nm/520 nm (excitation/emission).

### 2.4. Transendothelial Migration (TEM) Assay

Transwell culture systems with cell inserts (pore size 8.0 µm, BD Falcon) were employed for the TEM assay. Endothelial EA.hy926 cells (6 × 10^4^ cells) were seeded in fibronectin-coated cell inserts of 12-well transwell plates (upper chambers), whereas stromal HS-5 cells or FRC (1 × 10^5^ cells) were separately seeded in the 12-well transwell plates (lower chambers) at 2 days before coculture. One day before coculture, the EA.hy926 cells were stimulated with 10 ng/mL TNF-α overnight. The residual TNF-α was removed by extensive washing. Latently infected J1.1 and ACH-2 cells and acutely infected Jurkat and A3.01 cells (at 2 days post infection) were labeled with 2.0 µM CellTracker CMFDA. Uninfected Jurkat and A3.01 cells were similarly treated. The cells (1.2 × 10^6^ cells) were resuspended in 600 µL of RPMI1640 containing 10% FBS and 50 ng/mL chemokines, CCL19, CCL21, and CXCL12 and were added to the EA.hy926 cell monolayers in the cell inserts. The inserts were placed in 12-well transwell plates, where stromal cells (HS-5 or FRC) were grown and incubated in 1.5 mL of RPMI1640 containing 10% FBS and 50 ng/mL chemokines (CCL19, CCL21, and CXCL12). After 6 h of incubation at 37 °C, the cells were collected from the wells (lower chambers) and lysed with RIPA buffer. The MFI in the lysate was measured using a microplate reader at 480 nm/520 nm (excitation/emission).

### 2.5. In-Vitro Reconstitution of the Lymphatic Endothelium

The Collagen Gel Culturing Kit (Nitta Gelatin, Osaka, Japan) was used to reconstitute an extracellular matrix embedded with stromal cells, according to the manufacturer’s instructions. In brief, solution A (3 mg/mL collagen, Cellmatrix Type I-A), solution B (5× concentrated RPMI or DMEM), and solution C (reconstitution buffer containing 260 mM NaHCO_3_, 50 mM NaOH, and 200 mM HEPES) were mixed in a ratio of 7:2:1 on ice (reconstituted collagen solution). The reconstituted collagen solution was poured into 6-well plates and incubated at 37 °C for 30 min for solidification (base layer). HS-5 and HS-27A cells (1.5 × 10^6^ cells/well) were labeled with 5 µM CellTrace CFSE (Thermo Fisher) and were suspended in solution B. The 7:2:1 mixture of solution A, solution B (containing cells), and solution C was poured onto the base layer and solidified by incubation at 37 °C for 30 min (cell-collagen layer). After solidification, the culture medium was added and incubated at 37 °C for 1 week. Then, EA.hy926 cells (5 × 10^5^ cells/well) were labeled with 10 µM CellTracker CMTMR (Thermo Fisher) and seeded on the cell-collagen layer. The following day, EA.hy926 cells were stimulated with 10 ng/mL TNF-α overnight and washed extensively. Latently infected J1.1 and acutely infected Jurkat cells were labeled with 5 µg/mL Hoechst 33342 (Thermo Fisher). Uninfected Jurkat cells were used as controls. The cells were resuspended in RPMI1640 containing 10% FBS and 50 ng/mL CCL19, CCL21, and CXCL12, added to the EA.hy926 cell layer, and incubated at 37 °C for 3 h. After washing with PBS, the samples were observed with a fluorescence microscope (BZ-8000, Keyence, Osaka, Japan). The z-stack images were collected at 10 µm intervals and processed to generate 3-D reconstituted images using Fiji in ImageJ software.

### 2.6. Flow Cytometry (FCM)

Cells were incubated with mouse monoclonal antibodies (mAb) (10 μg/mL) in PBS containing 0.5% FBS and 0.1% NaN_3_ on ice for 30 min and subsequently with Alexa Fluor 488-conjugated anti-mouse IgG (Invitrogen, Thermo Fisher Scientific) on ice for 30 min. After washing, the cells were fixed with 4% paraformaldehyde (PFA) in PBS and subjected to FCM (FC500M, Beckman Coulter, Brea, CA, USA). A total of 100,000 events were processed for each sample. The following mouse mAb were used in this study: anti-L-selectin b (DREG-56, BioLegend), anti-LFA-2 (RPA-2.10, BioLegend), anti-very late antigen 4 (VLA-4) (HP2.1, Aviva Systems Biology, San Diego, CA, USA), anti-ICAM-1 (RR1/1, eBioscience, Thermo Fisher Scientific), anti-vascular cell adhesion molecule-1 (VCAM-1) (6G9, abcam), anti-platelet endothelial cell adhesion molecule-1 (PECAM-1) (WM59, BioLegend). Anti-LFA-1 and anti-LFA-3 mouse mAb [29] were obtained from Dr. Yokota (National Institute of Infectious Diseases, Tokyo, Japan).

### 2.7. Statistical Analysis

The statistical significance of differences was analyzed with Mann–Whitney U test. A *p* less than 0.05 was considered significant.

## 3. Results

### 3.1. Latently HIV-1-Infected T Cells Efficiently Adhered to Endothelial Cells

Since adhesion of lymphocytes to the endothelium is an early process in the TEM of lymphocytes and closely related to homing ability, the adhesive ability of T cells was investigated by an adhesion assay (Figure 1A). Latently infected and acutely infected Jurkat (Figure 1A, upper) and A3.01 (Figure 1A, lower) were labeled with CMFDA, added to the monolayers of human endothelial EA.hy926 cells, and incubated for 1 h. The adhesion efficiency was estimated by calculating the ratio of the MFI of adhered T cells to that of the total T cells (input). For both the Jurkat and A3.01 cell lineages, the adhesion ability of latently infected cells (J1.1 and ACH-2) to EA.hy926 was higher than that of uninfected and acutely infected cells (Figure 1A). The adhesion ability of acutely infected cells was comparable to that of uninfected cells. These results suggest that the adhesion ability of T cells to endothelial cells, at least for Jurkat and A3.01 cell lineages, was enhanced upon latent HIV-1 infection.

### 3.2. Downregulation of the Expression of Surface Adhesion Molecules in Latently Infected T Cells

Previous studies have shown that HIV-1 infection causes downregulation of CD4 and major histocompatibility complex (MHC)-I [30,31,32]. It is possible that the expression levels of cell adhesion molecules are changed upon HIV infection, thereby altering the adhesion efficiency of infected cells. Thus, the cell surface expression of cell adhesion molecules was investigated by FCM (Figure 1B). The molecules included L-selectin, LFA-1, LFA-2 (CD2), LFA-3, VLA-4, VCAM-1, and PECAM-1. L-selection and LFA-1 are key molecules for the TEM of lymphocytes in lymph nodes. In J1.1 cells, the expression of all tested cell adhesion molecules (L-selectin, LFA-1, LFA-2, and VLA-4) were downregulated compared to Jurkat cells. In acutely infected Jurkat, the expression of VLA-4 was reduced, but that of LFA-2 was significantly upregulated (Figure 1B, upper). Analysis of ACH-2 showed that the expression of VLA-4 was downregulated, whereas L-selectin and LFA-2 expression was slightly upregulated. In acutely infected A3.01 cells, the expression of LFA-1 and VLA-4 was downregulated (Figure 1B, middle). Endothelial EA.hy926 cells were positive for expression of ICAM-1, PECAM-1, and LFA-3 but negative for VCAM-1 (Figure 1B, lower). These results are inconsistent with the cell adhesion ability, suggesting that the increased adhesion efficiency of latently infected cells was not linked to the expression levels of the cell adhesion molecules tested here.

### 3.3. Latently Infected T Cells Efficiently Transmigrated across Endothelial Cells

Stromal cells in lymph nodes, such as FRC and FDC, secrete chemokines, CCL19, CCL21, and CXCL12, which attract lymphocytes through the chemokine-receptor interactions. It is also known that the stimulation of lymphocytes with these chemokines induces the integrin activation [14,33], leading to lymphocyte extravasation, indicating that chemokines and stromal cells synergistically promote lymphocyte transmigration. Thus, we examined the TEM of latently and acutely infected cells using transwell systems containing these chemokines (Figure 2A). CMFDA-labeled lymphocytes were added to the EA.hy926 monolayer grown in cell inserts and incubated in presence of CCL19, CCL21, and CXCL12 at 37 °C for 6 h. The efficiency of TEM was calculated as the ratio of the MFI of the migrated T cells to that of the total T cells (input). In both Jurkat (Figure 2B, upper) and A3.01 (Figure 2B, lower) cell lineages, latent J1.1 and ACH-2 cells showed slightly higher TEM efficiencies than their parental uninfected cells. In contrast, the TEM efficiencies of acutely infected Jurkat and A3.01 cells were similar to or slightly lower than those of their parental uninfected cells (Figure 2B).

To examine TEM using a cell-culture system that is more relevant to transmigration into lymph nodes, stromal cells, HS-5, or primary FRC were grown in plate wells, and the TEM of T cells across an EA.hy926 monolayer was similarly tested (Figure 2C). The TEM efficiency of T cells, both infected and uninfected, was slightly higher in the presence of stromal cells (Figure 2C). This enhancement was apparent in the presence of primary FRC. It should be noted that among the three Jurkat cell lineages (uninfected, acutely infected, and latently infected J1.1), the TEM of latent J1.1 cells significantly increased in the presence of FRC (Figure 2C). Together, these data indicate that, at least in this cell-coculture system, latently infected cells efficiently transmigrated across endothelial cells, and the TEM ability was further enhanced by the presence of stromal cells. Thus, it is possible that that latently infected cells are more attracted to the stromal cells.

### 3.4. Efficient TEM in In-Vitro Reconstructed Lymphatic Endothelium Systems Required Live Stromal Cells

Lymph node stromal cells produce collagen and form a 3-D network, which functions as a scaffold for lymphocyte migration in vivo [19,34]. To mimic the microenvironment in the lymph nodes, which is composed of HEV and stromal cells in collagen fibers, HS-5 and HS-27A cells were labeled with CFSE and were embedded in a collagen matrix containing CCL19, CCL21, and CXCL12. After the stromal cells formed a network structure in the collagen gel, CMTMR-labeled EA.hy926 cells were grown to a monolayer on the collagen gel, and Hoechst-labeled T cells were further added to the EA.hy926 monolayer (Figure 3A). After 3 h of incubation in the presence of the chemokines, the in-vitro reconstructed cell culture was fixed with 4% PFA, and the z-stack images were collected using a fluorescence microscope. When stromal cells (HS-5 and HS-27A) were embedded in the collagen gels, Jurkat cells, whether infected or not, efficiently migrated into the gels (Figure 3B). J1.1 and acutely infected Jurkat cells similarly transmigrated into the gels across the endothelial EA.hy926 layer.

To evaluate the efficiency of TEM, the number of migrated cells in the z-projection image from the top of a collagen gel to the bottom was calculated. When HS-5 cells were embedded, J1.1 showed slightly higher TEM efficiency, whereas acutely infected Jurkat showed slightly lower TEM efficiency than uninfected cells (Figure 3D). However, this was not the case when HS-27A cells were used, although the efficiency of TEM using HS-27A cells was generally higher than that using HS-5 (Figure 3D).

Interestingly, when stromal cells were absent in this reconstruction system, none of the Jurkat cell lineages transmigrated into the collagen gels (Figure 3C, upper, and Figure 3D). Even when the EA.hy926 monolayer was further absent, none of the Jurkat cell lineages transmigrated into the gels (Figure 3C, lower, and Figure 3D). Conversely, when EA.hy926 cells were absent, but the stromal cells were embedded, Jurkat cells transmigrated into the gels (Figure 3E, left). However, importantly, no transmigration occurred when PFA-fixed stromal cells were embedded in collagen gels (Figure 3E, right). These results suggest that, at least in this reconstructed system, TEM of T cells absolutely requires live stromal cells (Figure 3F), as the presences of chemokines, collagen, and endothelial cells alone was insufficient for TEM of T cells.

## 4. Discussion

Lymphocyte adhesion to HEV via cell-surface molecules is an early step in the lymphocyte homing cascade. Numerous studies have repeatedly shown that HIV-1 infection downregulates cell-surface expression of CD4 and MHC-I mostly due to the expression of HIV-1 Nef [30,31,32]. Some studies have also reported that HIV-1 strongly downregulates cell adhesion molecules, such as L-selectin and LFA-1 [35,36]. A recent proteomics analysis has further revealed the downregulation of >100 molecules involved in cell adhesion, lymphocyte activation, and membrane transport upon HIV-1 infection [37]. These studies were performed using acutely infected cells. In contrast, less is known about the modulation of cell-surface molecules in chronic or latent HIV-1 infections. One study has reported that latently HIV-1-infected T cells in humanized mice display apparently normal phenotypes in terms of cell-surface expression, such as expression of CD3, CD4, CD28, CD44, CD69, CXCR4, and MHC-I [38]. We used latently infected T-cell lines and acutely infected T-cell lines, both of which were derived from the same parental T-cell lines, and explored the expression of cell-adhesion molecules and their adhesion to endothelial cells. CD3 downregulation in latent J1.1 cell line has previously been reported [23]. Our FCM analysis revealed that L-selectin and LFA-1, the cell-adhesion molecules that regulate TEM, were downregulated in J1.1 cells, but the L-selectin expression in ACH-2 cells was slightly upregulated (Figure 1B). Nevertheless, both J1.1 and ACH-2 exhibited enhanced cell adhesion abilities compared to their parental cells (Figure 1A). These results clearly indicated no correlation between the adhesion ability to endothelial cells and the expression levels of these cell-adhesion molecules, suggesting that other molecules/mechanisms are involved in T-cell adhesion to endothelial cells, especially in latent infection. It is likely that the activated state of the adhesion molecules (e.g., the extended form of LFA-1) and the accumulation of the molecules to the cell–cell contact site are involved in the adhesion ability.

HIV-1 generally impairs the migration of infected T cells. HIV-infected T cells exhibited an elongated shape and reduced the motility in the lymph nodes of humanized mice [18]. This migratory deceleration is likely attributed to the expression of HIV-1 Nef. Numerous studies have shown that Nef impairs T-cell migration and chemotaxis [35,39]. More importantly, Nef expression has shown to inhibit lymph-node homing of murine lymphocytes in vivo [40]. Our study showed that acute HIV-1 infection impaired the TEM of Jurkat and A3.01 cells (Figure 2B). This is most likely due to Nef expression during acute infection.

We used latently infected J1.1 and ACH-2 cells and explored the efficiency of TEM in two in-vitro assay systems: a transwell system (Figure 2) and an in-vitro reconstructed system (Figure 3). In both systems, we found that latent infection did not impair the TEM of J1.1 (and ACH-2) cells, as their TEM abilities were comparable to or better than those of uninfected parental Jurkat and A3.01 cells (Figure 2B). The efficient TEM of J1.1 and ACH-2 may be due to the lack of Nef expression in latently HIV-1-infected cells. In the in-vitro reconstructed TEM system in which collagen matrices were present underneath endothelial cells, both latently and acutely infected cells could transmigrate into the collagen gels at levels similar to uninfected cells, showing that the differences were not significant (Figure 3B,D). A previous study reported that the velocity of T lymphocytes is somewhat reduced in high-density collagen matrices [40]. It is possible that the concentration of collagen used in this study may have suppressed the TEM ability of T cells, resulting in little difference in the TEM ability between latently infected and acutely infected cells.

Comparison of the two TEM assay systems showed that T cells transmigrated into the plate wells in the transwell system regardless of the absence of stromal cells in the lower chambers (Figure 2), whereas in the in-vitro reconstituted TEM systems, T cells did not transmigrate into the collagen gels when stromal cells were absent (Figure 3C,D). The TEM of T cells into the collagen gels was dependent on the presence of live stromal cells in the gels (Figure 3E,F). Since some chemokines (CXCL12, CCL19, and CCL21) were present in both TEM systems, other cytokines or soluble factors secreted from live stromal cells are likely required for the TEM of T cells in these in-vitro reconstituted systems. The in-vitro reconstituted TEM system with stromal cell-embedded collagen better mimics the microenvironment in the lymph nodes but requires microscopic analysis, whereas the transwell TEM system is more suitable for quantitative analysis. Thus, our TEM assay systems are only available for analysis on limited stages/aspects in lymph-node homing of infected cells but need to be further improved toward a better understanding the in-vivo TEM in lymph nodes. A more relevant analysis would be that, using primary cells, the TEM ability of latently infected cells is evaluated, if possible, in autologous cell-coculture system. Since stromal cells have been experimentally shown to reactivate HIV-1 in latently infected cells [41], the TEM ability of latently infected cells may lead to reversal of HIV-1 latency in vivo.

## Figures and Tables

**Figure 1 viruses-13-01589-f001:**
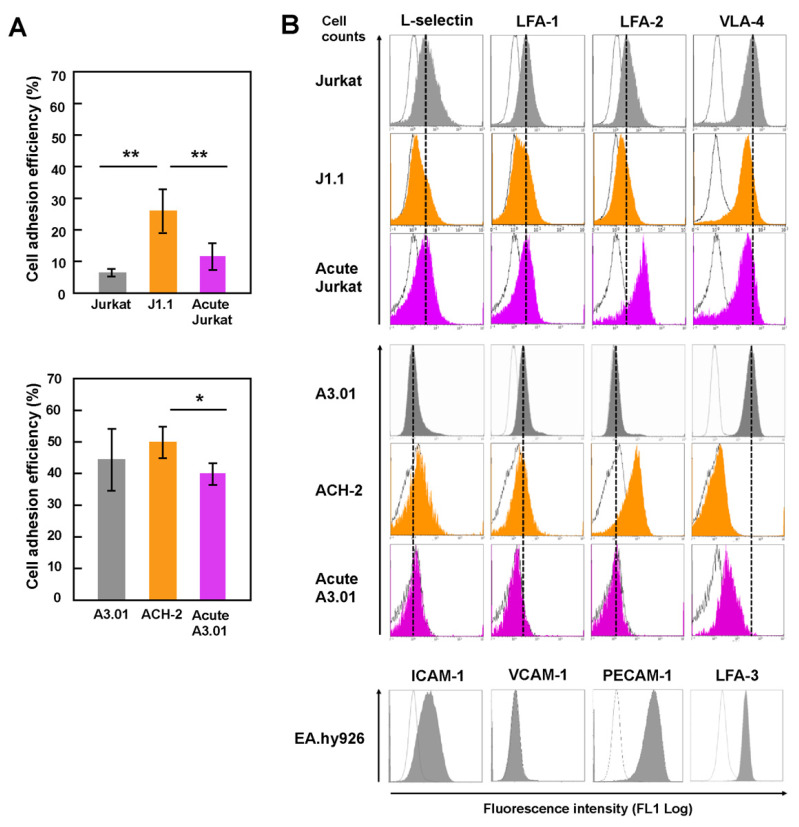
Cell adhesion ability and adhesion molecule expression of latently and acutely infected T cells. (**A**) Adhesion ability of Jurkat and A3.01 cell lineages. Uninfected, latently infected, and acutely infected cells of Jurkat (upper) and A3.01 (lower) cell lineages were labeled with CellTracker CMFDA. They were added to EA.hy926 cell monolayers and incubated for 1 h at 37 °C. After washing, the cells were lysed with RIPA buffer (1 mL). The MFI of the lysates (200 μL) was measured at 480 nm/520 nm and corrected using background control samples. Data are the mean ± S.D. (*n* = 15) from three independent experiments. * *p* < 0.05; ** *p* < 0.01. (**B**) Expression of cell adhesion molecules of Jurkat and A3.01 cell lineages. Uninfected cells, latently infected cells, and acutely infected cells of Jurkat and A3.01 cell lineages were immune-stained with anti-L-selection, anti-LFA-1, anti-LFA-2, and anti-VLA-4 mAbs and analyzed by FCM. EA.hy926 cells were immune-stained with anti-ICAM-1, anti-VCAM-1, anti-PECAM-1, and anti-LFA-3 mAb and subjected to FCM. White histograms show negative controls with secondary Ab alone. Dashed lines indicate peaks of their expressions in parental Jurkat and A3.01 cells.

**Figure 2 viruses-13-01589-f002:**
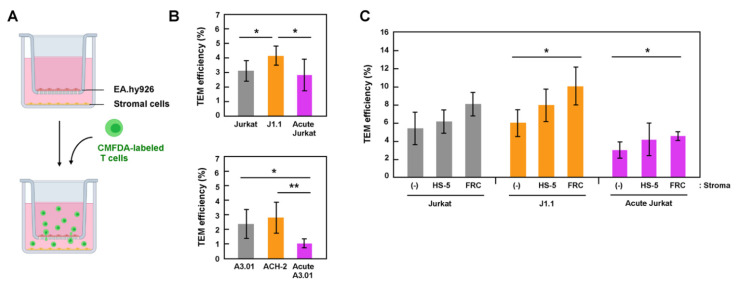
TEM efficiency of latently infected and acutely infected T cells in transwell systems. (**A**) A TEM system in transwell plates. EA.hy926 cells were grown in cell inserts of transwell plates (upper chambers), whereas stromal HS-5 or FRC were separately grown in plate wells (lower chambers). They were combined just before use. Uninfected, latently infected, and acutely infected T cells were labeled with CellTracker CMFDA. They were added to the EA.hy926 cell monolayers and incubated for 6 h at 37 °C in the presence of chemokines (CXCL12, CCL19, and CCL21). Transmigrated cells were collected and lysed with RIPA buffer (1 mL). The MFI of the lysates (200 μL) was measured at 480 nm/520 nm and corrected using background control samples. (**B**) TEM of Jurkat (upper) and A3.01 (lower) cell lineages across EA.hy926 cell monolayers. CMFDA-labeled T cells were added to the EA.hy926 cell monolayers in cell inserts, but stromal cells were absent in plate wells. Data are the mean ± S.D. from 4 or 5 independent experiments (**upper**) and 5 or 6 independent experiments (**lower**). * *p* < 0.05; ** *p* < 0.01. (**C**) TEM of Jurkat cell lineage (uninfected, latently infected, and acutely infected) through EA.hy926 cell monolayers in the absence (-) or presence of stromal cells (HS-5 and FRC) in plate wells. Data are the mean ± S.D. from at least three independent experiments. * *p* < 0.05.

**Figure 3 viruses-13-01589-f003:**
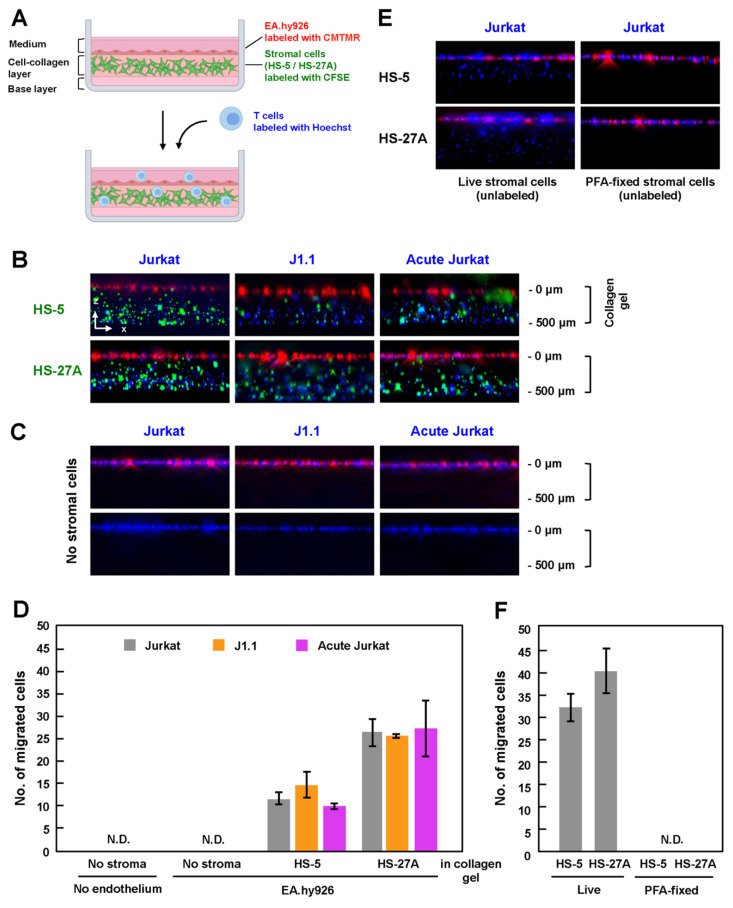
TEM of latently infected and acutely infected T cells in in-vitro reconstructed lymphatic endothelium systems. (**A**) In-vitro reconstituted lymphatic endothelium systems with collagen. Stromal cells were labeled with CellTrace CFSE and were embedded in reconstituted collagen gels with chemokines (CCL19, CCL21 and CXCL12). EA.hy926 cells were labeled with CellTracker CMTMR and grown to monolayers on the collagen gels. Uninfected Jurkat, J1.1, and acutely infected Jurkat cells were labeled with Hoechst 33342, added to the EA.hy926 monolayers, and incubated at 37 °C for 3 h. The z-stack images were collected and were processed to generate 3-D reconstituted images using Fiji in ImageJ software. (**B**,**C**) TEM of Jurkat cell lineage into the collagen gels. HS-5 and HS-27A were embedded (**B**) or not embedded (**C**) in the reconstituted collagen gels. EA.hy926 cells were present in ((**C**), upper) but absent in ((**C**), lower). (**E**) TEM of Jurkat cells into the collagen gels embedded with live (left) or PFA-fixed (right) stromal cells. The stromal cells were not labeled with CellTrace CFSE. (**D**,**F**) For quantification, the number of migrated cells was calculated in a z-projection image. Data are the mean ± S.D. from three independent experiments. N.D., not detectable.

## Data Availability

The raw data supporting the conclusions of this manuscript will be made available by the authors, without undue reservation. The data are available on request.

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
