# Peer review of "Efficient Transendothelial Migration of Latently HIV-1-Infected Cells"

_viruses, 2021, doi:10.3390/v13081589_

Round 1

Reviewer 1 Report

The manuscript by Tanabe et al, Efficient transendothelial migration of latently HIV-1-infected cells, investigate the migration capabilities of HIV-infected cell lines. The manuscript provides preliminary insights into the expression of cell adhesion molecules, and migration properties in chronically infected cells compared to acutely infected cells. The conclusions of the study are limited to the cell lines used in the study and further investigation in primary cells is required for a definitive conclusion. However, the conclusions presented in this work are supported by the results. Some critical concerns are summarized in my specific comments below.

  1. Authors should discuss the statistical analysis with a statistician. Non-parametric tests must be used.
  2. The source of the mAbs used for flow cytometry is unclear. Are these antibodies authenticated? Can authors provide references for the use of these antibodies?
  3. Authors should include uninfected and infected primary cells as controls for the expression of adhesion molecules comparing them to the transformed cell lines used in this study.
  4. Limitations of the study in terms of using exclusively cell lines, are lacking. How are these results comparable to primary latently infected cells?
  5. The discussion is somewhat repeating the results section. It should be focused on discussing the results.
  6. Lines 325-336: Authors discuss that the concentration of collagen used in the study may be too high. How was this concentration selected? Also, what are the conclusions of having the difference between both systems? What are the implications of this study for primary cells in people living with HIV?

Author Response

(Comments and suggestions by Reviewer 1)

The manuscript by Tanabe et al, Efficient transendothelial migration of latently HIV-1-infected cells, investigate the migration capabilities of HIV-infected cell lines. The manuscript provides preliminary insights into the expression of cell adhesion molecules, and migration properties in chronically infected cells compared to acutely infected cells. The conclusions of the study are limited to the cell lines used in the study and further investigation in primary cells is required for a definitive conclusion. However, the conclusions presented in this work are supported by the results. Some critical concerns are summarized in my specific comments below.

(R) We appreciate the reviewer’s comment that our results, although preliminary, support our conclusions. We have now amended the manuscript as below.

(1) Authors should discuss the statistical analysis with a statistician. Non-parametric tests must be used.

(R) Following the suggestion, we asked a scientist working on statistical analysis of clinical data and discussed appropriate methods of statistical analysis for our study. Then we have reviewed all the graph data and analyzed the significances with Mann-Whitney U test (non-parametric tests).

(2) The source of the mAbs used for flow cytometry is unclear. Are these antibodies authenticated? Can authors provide references for the use of these antibodies?

(R) We have added the reference of anti-LFA-1 and anti-LFA-3 antibodies and the clone numbers of the commercially available antibodies.

(3) Authors should include uninfected and infected primary cells as controls for the expression of adhesion molecules comparing them to the transformed cell lines used in this study.

(R) This study focused on the features of latently infected cells comparing to uninfected and acutely infected cells. We found it difficult to prepare these three cells from individual donors at the same time and did not use primary cells in this study. We speculate that the discrepancy between the expression levels of the adhesion molecules and the adhesive ability are not because we used the cell lines but possibly due to the activated state and/or the intracellular accumulation of the adhesion molecules. We have now added this discussion in the text.

(4) Limitations of the study in terms of using exclusively cell lines, are lacking. How are these results comparable to primary latently infected cells?

(R) We used cell lines throughout this study to compare latent infection, acute infection, and non-infection. However, we admit the reviewer’s critique and have added some text suggesting the limitations of our data and a discussion on possible TEM systems using primary cells in the discussion section.

(5) The discussion is somewhat repeating the results section. It should be focused on discussing the results.

(R) We first apologize for a duplicate description in the discussion section in our original manuscript (the test starting with “We used—” in the 1st paragraph and the text of the 3rd paragraph). We have eliminated the former text from the discussion.

Following the suggestions, we have added a discussion about the discrepancy between the expression levels of the adhesion molecules and the adhesion ability. We also added a discussion about the efficient TEM ability of latently infected cells compared with acutely infected cells.

(6) Lines 325-336: Authors discuss that the concentration of collagen used in the study may be too high. How was this concentration selected? Also, what are the conclusions of having the difference between both systems? What are the implications of this study for primary cells in people living with HIV?

(R) We used collagen at a concentration that is generally used for 3-dimensional organoid culture. The concentration (final 2.1 mg/mL) was in the middle between the high/low concentrations used in a study by Stolp et al.

We have compared the two in vitro TEM systems and have added an additional discussion (the advantages/disadvantages of the two systems) to the text. The conclusions on the differences of our data between the two systems cannot be made at present.

We agree that the implications of the TEM ability of infected cells in HIV-1 patients need to be suggested and have added a discussion to the last paragraph in the discussion section.

Reviewer 2 Report

Results sections have information related to the introduction and methods. Thus, it is challenging to understand the questions addressed.

If these can be filtered, it will increase readability.

Author Response

(Comments and suggestions by Reviewer 1)

Results sections have information related to the introduction and methods. Thus, it is challenging to understand the questions addressed.

If these can be filtered, it will increase readability.

(Responses)

We appreciate the reviewer to consider our introduction and results clearly described.

However, the reviewer finds that some text in the result section needs to be incorporated into the introduction section. We admit this critique and the explanations on the adhesion molecules and the chemokines produced stromal cells have been incorporated into the introduction section. We also removed the text on the methods from the result section. To make the text easily readable, we have added some text explaining why and how the experiments were performed instead.

In the discussion section, we added a discussion about the discrepancy between the expression levels of the adhesion molecules and the adhesion ability. We have also stated that our study only contributes to understand limited stages in lymph node homing of HIV-infected cells.

We apologize for a duplicate description in the discussion section in our original manuscript (the test starting with “We used—” in the 1st paragraph and the text of the 3rd paragraph). We have eliminated the former text from the discussion.

Reviewer 3 Report

The question addressed by tjis study is relevant, however the methods used allowed only small progress in understanding the process, so it must be considered of limited importance, since no clarification on the molecules involved in TEM, not the action of stromal cells has been obtained. Within such limits, the study is a good exercise with adequate controls and some suggestions for future experiments able to solve the issue. Severe limitations should be underlined in the discussion, hinting at the preliminary mature of these findings. There are some typos in lines 81-83 (abilities to... and system).

Author Response

(Comments and suggestions by Reviewer 2)

The question addressed by tjis study is relevant, however the methods used allowed only small progress in understanding the process, so it must be considered of limited importance, since no clarification on the molecules involved in TEM, not the action of stromal cells has been obtained. Within such limits, the study is a good exercise with adequate controls and some suggestions for future experiments able to solve the issue. Severe limitations should be underlined in the discussion, hinting at the preliminary mature of these findings. There are some typos in lines 81-83 (abilities to... and system).

(Responses)

We appreciate the reviewer to consider our study relevant and a good lesson with adequate controls.

However, the reviewer finds that our in vitro TEM systems only allow a small contribution of our understanding. We admit this critique and have stated in the discussion section that our study only contributes to understand limited stages in lymph node homing of infected cells. We have also added a discussion about the discrepancy between the expression levels of the adhesion molecules and the adhesion ability, although we still cannot clarify the mechanisms of TEM of latently infected cells.

The explanations on the adhesion molecules and the chemokines produced stromal cells have been removed from the result section and incorporated into the introduction section. To make the text easily readable, we added some text explaining why and how the experiments were performed instead.

We apologize for a duplicate description in the discussion section in our original manuscript (the test starting with “We used—” in the 1st paragraph and the text of the 3rd paragraph). We have eliminated the former text from the discussion.

The typos pointed out by the reviewer have been corrected.

Round 2

Reviewer 1 Report

Thank you for addressing the comments. The manuscript is now suitable for publication.